# Benefits and Challenges of Video Consulting for Mental Health Diagnosis and Follow-Up: A Qualitative Study in Community Care

**DOI:** 10.3390/ijerph20032595

**Published:** 2023-01-31

**Authors:** Yusuf Sheikh, Ayesha Ali, Aya Khasati, Alan Hasanic, Urvi Bihani, Raja Ohri, Keerthi Muthukumar, James Barlow

**Affiliations:** Imperial College Business School, Imperial College London, London SW7 2AZ, UK

**Keywords:** mental health, telemedicine, general practice

## Abstract

Mental health services continue to experience rising demand that exceeds capacity. The COVID-19 pandemic exacerbated this crisis, with access to services being reduced. Although video consultations (VCs) are a solution, usage in UK community mental healthcare settings remains limited. This study aims to investigate psychiatrists’ and general practitioners’ (GPs) perceptions of the benefits and challenges of VC for the diagnosis and follow-up of general adult mental health patients in the community during the COVID-19 pandemic. Semi-structured interviews in NHS community mental healthcare settings were conducted. Psychiatrists (*n* = 11) and GPs (*n* = 12) were recruited through purposive sampling. An explorative qualitative approach was employed. Data were analysed using thematic analysis. Four key themes were identified: (1) patient access to VC, (2) suitability of VC for mental health consultations, (3) information gathering with VC and (4) clinician satisfaction with VC. This study provides valuable insights into the experiences of psychiatrists and GPs working in the UK during the COVID-19 pandemic. To facilitate a digital-first future for the NHS, greater investment in remote technologies is required, particularly in the context of growing mental healthcare demand. Though face-to-face consultations remain the gold standard, VC provides an efficient way of communicating with patients, particularly those with less severe forms of mental illness.

## 1. Introduction

Psychiatrists today are caught in the crossfire of two pandemics: the incumbent COVID-19 pandemic, and an emerging mental health (MH) crisis. It became immediately apparent after the first lockdown began in March 2020 that the pandemic and its consequences were going to take a considerable toll on the population’s mental health. In addition to the global excess mortality, the pandemic has had a significant impact on mental illness; a recent Lancet study estimated a global increase of 25.6% in anxiety disorder cases as a result [1]. Indeed, the estimated impact of COVID-19 on MH services suggests an almost 40% increase in demand by 2023, with the greatest demand seen in primary care (53%) [2].

The unexpected onset of the COVID-19 pandemic made virtual care an essential means of care provision; the use of video consultations (VCs) has become widespread across clinical specialties [3]. Video consulting is a well-established medium for the routine provision of healthcare, with international use reported across a range of care settings and clinical specialties, including MH and primary care settings [4,5,6]. Recent studies have found VC to be an effective consultation modality for conditions that do not necessitate physical examination, such as mental illness [7,8]. Despite clear advantages over other forms of remote consultations (RCs), such as telephone consultations [9,10], and the efforts of the most recent NHS England Long Term Plan (2019) to mandate the availability of VC within the next few years [11], uptake in UK community mental healthcare settings prior to the pandemic was low [12]. 

With the Royal College of General Practitioners and other similar bodies stating that RC will form a fundamental role in future care delivery, the perspectives of psychiatrists and general practitioners (GPs) towards this technological application is a significant determinant of the future role of VC [13,14]. It is imperative that the experiences of healthcare professionals whilst using RC technology during the pandemic are evaluated so that challenges can be addressed moving forwards.

To date, while recent studies have provided insight into the usage of VC internationally, literature findings lack applicability to mental health physicians working in UK community settings [15]. Few UK studies have been published exploring staff attitudes towards VC in mental healthcare settings [16], and none to our knowledge focus specifically on its use in community mental healthcare for diagnosis and follow-up. Through an explorative qualitative design, this study aims to explore clinician perspectives on the benefits and challenges in using video consultations for general adult mental health diagnosis and follow-up in the community during the COVID-19 pandemic.

## 2. Materials and Methods

The present study reports this research using the Standards for Reporting Qualitative Research framework [17]. Data was collected from clinicians using semi-structured interviews (SSIs). The authors assert that all procedures contributing to this work comply with the ethical standards of the relevant national and institutional committees on human experimentation and with the Helsinki Declaration of 1975, as revised in 2008. All procedures involving human subjects/patients were approved by the Imperial College Research Ethics Committee on 08/02/21 (ICREC Reference: 20IC6508).

### 2.1. Study Design

This study used a qualitative exploratory design through SSIs to explore lived experiences of physicians providing MH care remotely in the community during the COVID-19 pandemic. SSIs were directed by a flexible interview guide and supplemented with follow-up questions to allow for a balance between focus and breadth of content [18].

Interviews aimed to cover physician-perceived benefits and challenges of using VC during the pandemic for mental health patients in the community. Questions were open-ended to allow in-depth responses. Themes identified from the existing literature facilitated an initial topic guide, which was then refined through pilot interviews and developed iteratively as interviews convened [7,19].

### 2.2. Sampling and Recruitment

Although MH care provision encompasses a variety of healthcare professionals, diagnosis and follow-up are primarily the responsibilities of general adult psychiatrists and GPs [20]. Therefore, eleven psychiatrists and twelve GPs were recruited to participate. The inclusion and exclusion criteria for participant selection are summarised in Table 1.

Participants meeting the eligibility criteria were recruited via LinkedIn, personal contacts and professional social media groups. In this study, maximum-variation purposive sampling, a type of nonprobability sampling, was used to ensure diversity in ages, sex, ethnicity, location and specialty [21]. An optional demographic questionnaire was also disseminated to participants.

In qualitative studies, it is acknowledged that the focus is not on statistical representativeness, but rather achieving depth of understanding [22]. Consequently, the sample size was dictated by thematic saturation [23]. This was reached at *n* = 23 and determined following an iterative process whereby initial SSIs were analysed to inform further data collection.

In total, 23 physicians (11 community general adult consultant psychiatrists and 12 GPs) participated in the interviews. The mean and median ages for this study were 49.7 and 48 years, respectively (95% CI 43.9–55.6 years). Professional characteristics are displayed in Table 2.

### 2.3. Data Collection

All interviews were conducted by the same two authors (YS and UB) using the Microsoft Teams video conferencing platform between April and May 2021. A participant information sheet was provided to interviewees and written informed consent was obtained prior to the interview. Interviews were audio-recorded, anonymised and stored on a secure drive for further analysis. Depending on subject availability, interviews lasted between 25 and 60 min.

### 2.4. Data Analysis

Interviews were transcribed verbatim using Otter.ai, a secure online transcribing software [24]. Transcripts were checked against audio files for accuracy and then immediately destroyed. Thematic analysis was utilised, applying Braun and Clarke’s six-step framework [25], which employs a theoretically flexible inductive approach to analysing qualitative data.

All transcripts were double-coded inductively to enable data familiarisation (phase one), verify accuracy and maximise rigour [26]. In phase two, a preliminary codebook was compiled from seven transcripts; these codes formed the majority of the finalised codebook. Codes were then sorted under four broader themes to form an initial thematic map in phase three. The remaining 16 transcripts were coded, and disagreement tackled through discussion. Themes were then reviewed and collated (phase four), before being defined and named in phase five. Finally, the report was produced, outlining the study findings (phase 6).

## 3. Results

The participant interviews revealed insights into the benefits and challenges experienced by GPs and psychiatrists. Following analysis of the interview data, four themes (alongside their respective subthemes) were identified. These four themes were patient access to VC; suitability of VC for mental health consultations; information gathering with VC; and clinician satisfaction with VC (Table 3).

### 3.1. Theme 1: Patient Access to VC

Some benefits and challenges discussed by participants were related to the inherent characteristics of the remote medium of video consultation. It was frequently mentioned that VC had potential to both improve and hinder patient access to clinical services, depending on patient demographics. 

#### 3.1.1. Improved Accessibility

Physicians commonly cited improved access for younger and working demographics. Comparatively, it was easier for younger patients to share information via VC than FTF. VC was especially reassuring for younger patients with privacy concerns or associated stigma: “I think it’s easy to be less stigmatising, not feeling that they’re going to a place that says a mental health unit … for the younger people it’s been easier for them” (01 CP).

Most clinicians perceived VC to be more convenient for patients, due to time and cost savings. This was seen as especially beneficial for working patients: “getting appointments is easier for the patient, they don’t have to drive from work or come from wherever they are. Distance, parking, traffic—overall on the whole that is easier for the patients as they are at work, and they just get a phone call from us so that way it is easier” (03 GP).

VC also increased access to patients living far away from the practice or MH patients who found it difficult to attend FTF appointments due to a lack of motivation, increased anxiety or abnormal sleep patterns. The increased reach, increased attendance rates and ease of appointment rescheduling afforded by VC led to widespread reductions in “Did Not Attend” (DNA) appointment rates among MH patients: “a lot of mental health patients find it very hard to be motivated… with some of our depressed patients who find it very hard to leave the house, you’re probably more likely to get them on the phone” (11 GP); “So the DNA rate is significantly low[er] [with VC], it’s brilliant. And so there is a feelgood factor to that, you know, you come to work and you think, okay, I have eight patients booked today, all eight patients have attended and interacted” (03 CP). 

Unsurprisingly, a key benefit of VC was its role in preventing COVID-19 transmission. During a period of travel and social-distancing restrictions, this helped to facilitate remote MH care access during the pandemic, enabling continuity of care: “in the pandemic, we all need to be aware of the health hazards of bringing people to the surgery” (06 GP).

#### 3.1.2. Reduced Accessibility

Most GPs and psychiatrists reported that older adult populations were particularly disadvantaged by the shift to RC. This was especially prevalent with VC, a consequence of both unfamiliarity with video technology and decreased access to digital equipment: “Well the older adults often don’t have a mobile with video, so often video for them is quite difficult” (09 GP).

Furthermore, many participants felt that VC may exacerbate existing health inequalities through reduced access to less digitally equipped patients. The relevance of this digital divide was greatly emphasised for patients from poorer socioeconomic backgrounds: “they [patients] have said there’s no money for even a call, for them to call us” (03 GP); “patients who are isolated, who barely even open the post, forget you know having access to anything technical or being tech savvy in any way. So the majority don’t have smartphones” (02 CP).

### 3.2. Theme 2: Suitability of VC for Mental Health Consultations

Clinicians agreed that specific types of consultations and MH conditions were better suited to VC, whilst the remainder should be conducted in-person. 

#### 3.2.1. Suitable for Monitoring and Follow-Up

Both GPs and psychiatrists felt that VC was suitable for follow-up consultations, which often involved medication reviews and further risk assessments. However, video conferencing appeared to offer little additional benefit compared to telephone conferencing: “If you’ve already seen the patient once face-to-face for an assessment and you’re merely following them to check on medication or further assess risk… telephone consultation often suffices” (09 CP). 

Importantly, the benefit of VC in facilitating virtual home visits was particularly relevant for psychiatrists: “For home visits, it [VC] saves time. If I had to do a home visit, [I’d do] maybe two in half a day. Now I can see four people in that slot” (06 CP).

#### 3.2.2. Unsuitable for Initial Assessment and Crisis Management

Clinicians reported that initial MH consultations were challenging over video, as it was difficult to "get a sense of the person”, thus affecting their ability to make an initial diagnosis. Although there was agreement that VC had a significant role in the follow-up of MH patients, many clinicians stated that initial consultations needed to be conducted in-person: “Certainly, for new patients, people you’ve not seen before… very early on in the assessment process, you’d want to do a face-to-face consultation” (05 CP).

Several clinicians highlighted that VC was ineffective for sectioning patients, which is defined as the involuntary detainment and treatment of MH patients under the Mental Health Act (NHS, 1983): “Patients need to be seen face to face in urgent situations… for Mental Health Act type interventions” (09 CP).

GPs and psychiatrists also stated that VC was ineffective for emergency interventions and crisis management due to the difficulty in de-escalating conflict remotely. Consequently, clinicians felt these situations required transition from VC, and were better-suited to FTF: “She [the patient] came in and broke a window in reception and had a panic attack… eventually we calmed her down… I just don’t think we’d have done that on the phone. One of us would have hung up in frustration” (11 GP).

#### 3.2.3. Suitable for Certain Mental Health Conditions

Participants widely acknowledged that VC was beneficial in improving access and engagement for patients with specific mental health conditions, including mild-to-moderate depression and anxiety, OCD and agoraphobia. Notably, several clinicians felt that VC may have benefits over FTF for patients with mild-to-moderate depression and anxiety: “She already suffered anxiety for many years… she would not allow anyone into her house, and she would not dare to come out of the house… for this kind of thing telehealth has helped” (03 GP).

#### 3.2.4. Unsuitable for Certain Mental Health Conditions

Both GPs and psychiatrists reported RC as being unsuitable for the diagnosis and follow-up of severe mental illness, suggesting that these patients were more difficult to assess remotely. One psychiatrist stated that the assessment of the severity of illness was hindered: “It’s a question of being able to assess the severity… I don’t think you can accurately assess [that] on a video link” (11 CP).

VC was deemed particularly unsuitable for psychotic patients. This is fundamentally due to paranoia, suspicious thoughts and generalised delusions regarding the technology used for the consultation: “Paranoid patients, people with psychosis, can get paranoid thoughts that somebody is trying to record them” (02 GP).

Additionally, assessment and management of MH patients requiring physical examination, such as those with eating disorders, were especially difficult to do remotely. Importantly, clinicians believed VC to be unsuitable for the management of suicidal patients, due to uncertainty regarding patient safety and location: “If you’ve got someone in your room that’s suicidal, you know where they are and you can contain them in the room…whereas if they’re ringing in, you’ve got no idea” (11 GP).

### 3.3. Theme 3: Information Gathering with VC

GPs and psychiatrists generally benefited from the presence of visual cues and insight into patients’ home environments. However, inappropriate consultation environments, privacy concerns and impaired doctor–patient rapport remain ongoing challenges with VC.

#### 3.3.1. Presence of Visual Cues

Both GPs and psychiatrists described the presence of visual and non-verbal cues over video as beneficial to their diagnostic confidence. For mental illness, most physicians felt that the combination of the visual component afforded with VC and the histories provided were sufficient for both diagnosis and follow-up: “With mental health conditions, more or less just with the history, we are able to get the diagnosis… diagnosis is not a problem [with video]” (02 GP).

The use of video enabled appearance and grooming cues to be detected, with the visual element seen as a key benefit compared to previous teleconsulting modalities: “How is someone looking after themselves?… You can kind of tell that by what they’re wearing, which you can’t tell on the phone, but you could tell on a video” (11 GP).

Importantly, interviewees also reported increased information sharing over video, aiding diagnostics. This was achieved through greater family involvement, online resource sharing and insight into the patient’s home environment: “If they consent, then we can also talk to their family and get some more information, because they’re all at home in the same place” (04 CP); “If you’re doing an assessment, the person can show you a full room shot… if you use it [VC] in a dynamic way, you get all the advantages of a home visit without anyone having to leave home” (10 CP).

Several clinicians felt that their patients demonstrated greater engagement and improved focus with VC, with the added distance and home comfort leading to patients opening up more: “because they’re sitting at home, they will sort of detail everything, which is good for us… openness and transparency does help a psychiatric diagnosis and even the therapeutic rapport, you know, which is important” (06 CP).

#### 3.3.2. Disruptive Consultation Environments

Disruptive patient environments were highlighted as a key challenge with VC. Clinicians stated that some patients were unable to obtain private spaces to conduct the consultation or did not recognise the need to do so. Factors such as busy households, interruptions from children and taking calls on public transport were especially distracting, often resulting in a need to reschedule appointments: “it was very difficult to have a focused conversation, which would never be a problem face-to-face” (02 CP); “They said they can’t continue because they don’t have a private place to talk and there are people in the house.” (08 CP).

Unsuitable consultation environments led to expressions of concern surrounding both confidentiality and safeguarding with VC, compared to FTF. Clinicians stated that they could never be completely sure who else was present during remote consultations, and concerns regarding coercion could also be detrimental to information gathering. This was of particular concern with vulnerable patients, such as those who may experience domestic abuse: “there is a risk that someone is either being coerced or abused and you can’t be sure the coercer or abuser isn’t listening in” (10 CP).

Furthermore, GPs and psychiatrists felt that the doctor’s physical presence with in-person consultations was reassuring to patients. The loss of this therapeutic rapport over video was perceived as disadvantageous: “sometimes it’s that reassurance that you’re seeing a professional… you don’t see the person and you kind of lose that human contact, even in video” (09 GP).

### 3.4. Theme 4: Clinician Satisfaction with VC

GPs and psychiatrists reported mixed satisfaction with VC. Some benefits and challenges experienced were related to the video consultation technology and experience of working during a pandemic, rather than the consultation itself.

#### 3.4.1. Improved Convenience

Clinicians highlighted the benefit of VC in providing opportunities to work from home, which were especially important for staff with childcare responsibilities. Many participants acknowledged improved job satisfaction, with one GP stating that they felt “happy and confident” managing patients over video. The combination of working from home and the reduced need to commute contributed to a better work–life balance: “There’s been a balance between being in the office and being at home… you’re spending less time travelling… it’s generally been beneficial for the working pattern” (09 CP).

Many participants experienced both greater flexibility and increased efficiency throughout their working day. This enabled clinicians to take on additional responsibilities, attend teaching sessions and use their time more effectively: “It has meant that you can juggle lots of different things. So you could do a clinic in the morning … you might even go on to do a teaching session…it’s opened up a whole range of different things that you can do with your time, not just seeing patients” (05 CP). “There’s no time wasting. When I say 12:30, CPA to start, I join around 12:28, everybody’s on time, it’s brilliant” (03 CP).

Importantly, VC was also perceived to improve clinician safety, especially amongst psychiatrists attending remote community MH team ward rounds. The increased distance between patients and physicians over video was seen as less “risky” and prevented the need for additional precautions to ensure clinician safety. This was particularly relevant regarding aggressive and agitated MH patients: “I’ve been to some ward rounds remotely… if the patient’s getting a bit more agitated or unsettled, we don’t have to stand up or move… it’s quite helpful in some of those aspects to be distanced from maybe more aggressive patients” (01 CP).

#### 3.4.2. Technological Challenges

A lack of training was commonly emphasised as a challenge to VC usage, with clinicians expressing inexperience with both the technology itself, and how to structure remote MH consultations. Physicians expressed a need for further guidance to help them overcome the unfamiliarity of the system when navigating VC platforms, especially with video setup and troubleshooting: “People are less comfortable with RC… there needs to be training for professionals on how to use video and telephone consultations, how to set things up, how to start it, how to end it” (05 CP). “There is a different set of communication skills you need for audio and video consultations… everybody’s just doing it their own way… I think there needs to be certain parameters set” (08 CP).

Technical issues with VC platforms were also frequently experienced. Poor connectivity was reported as disruptive to the consultation flow as neither party could hear or see each other clearly. Clinicians stated that they often needed to change over from a VC to a telephone consultation or FTF to continue the consultation: “Some days the video is not working and… the photo is not there. So there have been a lot of technology issues so ultimately we abandon the video call” (03 GP).

#### 3.4.3. Staff Resistance

Both GPs and psychiatrists spoke about staff resistance to the increased use of RC, both over video and telephone. This clinician resistance can be attributed to discomfort with technology and generalised resistance to organisational change. One GP described their colleagues as technophobic, whilst others expressed discontent, specifically with VC: “[There is] fear by staff, who don’t like seeing themselves on screen and listening to themselves on screen” (10 CP).

However, several physicians stated that their resistance to using video consultations was a result of their reluctance to change, as opposed to resistance with video technology itself: “It’s just my view… I’m getting on in years, I’ve been set in my ways, and maybe I’m resistant to change” (11 CP).

## 4. Discussion

### 4.1. Summary of Findings

This qualitative exploratory study provides a rich overview of the benefits and challenges experienced by GPs and psychiatrists whilst using VC during the pandemic for mental health consultations in the community. The findings of our study have been summarised into four themes: patient access to VC, suitability of VC for mental health consultations, information gathering with VC and clinician satisfaction with VC.

Several physicians highlighted that both younger and working populations benefited from VC, as it helps to reduce mental-health-associated stigma while providing savings in cost and time travelled. Meanwhile, older populations were often disadvantaged, due to unfamiliarity with the video technology. 

VC was deemed suitable for follow-up consultations such as medication reviews. However, difficulties in establishing initial patient rapport and de-escalating conflict meant that first consultations and crisis management needed in-person consultations.

The presence of visual cues and the ability to gain insight into a patient’s home environment with VC were beneficial to diagnosis and helped facilitate information gathering. However, inappropriate consultation environments led some physicians to express safeguarding and confidentiality concerns.

Overall, clinician satisfaction was mixed. Time savings, convenience and the flexibility in working from home were all positives. However, physicians reported frustration with video technology, lack of training and generalised resistance to organisational change.

### 4.2. Strengths and Limitations

To our knowledge, this qualitative study is the most in-depth investigation of clinician experiences with VC for mental health diagnosis and follow-up in UK community settings. These findings may be generalisable to other national health systems and are likely to be of general interest to psychiatrists, healthcare policymakers and the UK population. Nevertheless, it is important to acknowledge several study limitations.

Non-random recruitment of participants via purposive sampling techniques meant interview findings are limited by selection bias [27]. Although participants were selected opportunistically to ensure a balanced gender/specialty split and reach thematic saturation, the insufficient representation of trainees, physicians in rural areas (*n* = 1) and those operating in certain regions in the UK (e.g., West Midlands) limits transferability.

We are cognisant of the timing of our research in the context of the pandemic, and as such, COVID-19 remains a confounding variable. Although the study explicitly focuses on experiences with VC, clinicians may have had difficulty isolating VC experiences from those pertaining to system-wide challenges during the pandemic. For example, interviewees reported staff resistance and burnout, though recent findings describe a heavy toll of COVID-19 on physicians irrespective of VC [28].

### 4.3. Comparison with the Existing Literature

The benefits of VC reported were similar to those identified in international studies before and during COVID-19 [4,7,29]. In alignment with studies on patient perspectives, clinicians felt that patient satisfaction with VC for mental illness was high [30]. For younger, working patients, and those suffering from depression, anxiety and agoraphobia, VC was particularly convenient and time-saving, supporting findings from previous studies [8,16,31]. Our study revealed that clinicians also found VC to be time-saving, improving flexibility in otherwise tightly organised working days.

Studies have shown that the introduction of VC raises concern about clinical quality, confidentiality, safeguarding and accountability [32,33]. Clinicians in the present study were generally positive about VC’s implications for mental illness, yet expressed concerns over privacy, safeguarding and whether RC would be the default post-pandemic. This was of particular concern during consultations with mental health patients experiencing domestic abuse, a finding that has been echoed in previous studies [19]. Offering a confidential space to address sensitive topics may be more difficult via VC, which may lead to important biopsychosocial concerns being missed. Patient-determined consultation times and provision of patient preparation materials prior to the consultation may mitigate this problem in the future.

Additionally, older populations were commonly identified by participants as less suited to VC. This challenge has been well-documented in the telehealth literature and may be more pronounced with video consultations as telephone technology is more familiar and popular amongst older adult patients [34]. Previous studies have also suggested that VC may widen the digital divide and exacerbate existing healthcare disparities for those who may not have access to equipment such as computers or webcams to perform videoconferencing [35,36]. The exacerbation of pre-existing socioeconomic inequalities was also highlighted in our study. This challenge is especially pertinent to mental healthcare as social deprivation is a key risk factor for mental health problems [37].

As with previous studies, technical issues were frequent, and when experienced, clinician engagement and satisfaction with video decreased [5,38]. Physicians expressed the need for further guidance for clinicians and trusts, and training specific to remote MH consultations before VC becomes a mainstay in care provision.

### 4.4. Implications for Clinical Practice and Health Policy

The NHS Long Term Plan advocates for the adoption of digital tools, with plans to offer a “digital-first” option within the next 10 years [11]. Although findings from the present study suggest a clinician perception of patient satisfaction with video, further research is essential in determining cost effectiveness and identifying which MH patients will benefit from VC. Furthermore, the findings of this study necessitate additional remote MH-specific guidance to ensure optimal usage. The technical issues experienced reinforce the call for further investment into digital infrastructure and trust-wide training [39]. Through exploring clinician experiences, this study lays out the key benefits and challenges in using VC and provides a foundation for policymakers, psychiatrists and researchers to improve UK video-based consulting within community mental healthcare provision.

## 5. Conclusions

This study provides valuable insights into the experiences of psychiatrists and GPs working in the UK during the COVID-19 pandemic. Moving forward, face-to-face consultations will remain the gold standard. However, VC can provide a convenient and efficient way of communicating with patients, particularly during follow-up and for patients with less severe forms of mental illness. To enable a digital-first future for the NHS, greater investment in remote technologies, infrastructure and training is required, particularly for psychiatrists and GPs, in the context of the growing mental healthcare demand.

## Figures and Tables

**Table 1 ijerph-20-02595-t001:** Criteria for interview participation.

Inclusion Criteria	Exclusion Criteria
- Psychiatrists and GPs working in - community settings	- Healthcare professionals in - other specialties
- Works within the NHS - RC experience	- Works solely in private practice of other healthcare systems abroad - No RC experience

**Table 2 ijerph-20-02595-t002:** Professional characteristics of psychiatrists and GPs interviewed.

Specialty	Gender	Practice Region	Description of Practice Location
Psychiatrist	Female	East of England	Urban
Psychiatrist	Female	East of England	Urban
Psychiatrist	Female	London	Urban
Psychiatrist	Male	London	Urban
Psychiatrist	Male	North West	Urban
Psychiatrist	Prefer not to say	North West	Urban
Psychiatrist	Female	London	Urban
Psychiatrist	Male	London	Urban
Psychiatrist	Female	East Midlands	Suburban
Psychiatrist	Male	London	Urban
GP	Female	London	Urban
GP	Male	London	Urban
GP	Male	South East	Urban
GP	Female	London	Suburban
GP	Male	Yorkshire and the Humber	Suburban
GP	Female	Yorkshire and the Humber	Urban
GP	Male	East Midlands	Urban
GP	Male	Yorkshire and the Humber	Urban
GP	Male	North East	Urban
GP	Female	Yorkshire and the Humber	Suburban
GP	Female	Yorkshire and the Humber	Urban
GP	Female	East of England	nil

Note: one participant declined to fill out the demographic questionnaire.

**Table 3 ijerph-20-02595-t003:** Description of the themes and subthemes.

Themes	Subthemes
Theme 1: Patient access to VC	- Improved accessibility - Reduced accessibility
Theme 2: Suitability of VC for mental health consultations	- Suitable for monitoring and follow-up - Unsuitable for initial assessment and crisis management - Suitable for certain mental health conditions - Unsuitable for certain mental health conditions
Theme 3: Information gathering with VC	- Presence of visual cues - Disruptive consultation environments
Theme 4: Clinician satisfaction with VC	- Improved convenience - Technological challenges - Staff resistance

## Data Availability

The data presented in this study can be made available on request by the corresponding author.

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
