# Peer review of "Benefits and Challenges of Video Consulting for Mental Health Diagnosis and Follow-Up: A Qualitative Study in Community Care"

_ijerph, 2023, doi:10.3390/ijerph20032595_

Round 1

Reviewer 1 Report

The paper is well structured and easy to read. Background, study methodology, results and implication are clearly introduced.

There are several obvious weaknesses that can be improved:

I. The research questions in this paper are not very clear, and it is not easy to see the clear relationship between the thematic approach and the research questions. From the subject of the paper, it seems that the research questions are the benefits and challenges of the video consulting (“This study aims to investigate psychiatrists’ and general practitioners’ perceptions of the benefits and challenges of VC”[L15–16]); in the theme framework, the “benefits and challenges” are placed at the level of sub-themes or initial codes. Each theme should be clearly linked to the research objectives or research questions. Thematic approach should be able to clearly correspond to or answer relevant research questions.

II.The analysis or result section needs more clarification.

1. The current three themes and most of their sub-themes identified in the paper seem to belong to three topic areas and their subtopics. If these themes are not internally coherent, consistent, and distinct, they need to be further sorted, organized and defined;

2. The sub-theme levels and naming standards of the three themes are not well coordinated. For example, the two-sub themes of the first theme "access to VC" are "benefits" and "challenges"; The second theme, "suitability of a MH consultation to video", contains two types of the coded extracts—"benefits" and "challenges" under the four sub-themes; Among the three sub-themes under the Theme 3, "clinician satisfaction with conducting VCs", the hierarchical sort method of the first two sub-themes is the same as that of the sub-themes of Theme 2, while the third sub-theme "Staff resistance", seems to belong to the code category under the "challenge" category. “Challenges” are placed in different hierarchies as sub-themes or as the coded extracts.

3. Some overlaps between themes or sub-themes need to be re-examined and confirmed. The three themes of this paper have different or overlapping objects (people). The Theme 1 is for patients, the Theme 2 is for clinicians and patients at the same time, and the Theme 3 is for clinicians; Theme 1 (sub-theme 1) deals with the access convenience of patients, while Theme 2 (sub-theme 1) and Theme 3 (sub-theme 1) deal with the similar convenience of clinicians, etc.

III. Some parts for improvement in the paper

1. It is not appropriate to take the demographic characteristics of the participants as the results or findings of the thematic analysis, which can be placed in the section of 2.2. Sampling and recruitment. Can the two tables (Table 2 and Table 3) be combined? It could reflect the professional background described. Can the participants’ codes be included in the table?

2. The literature review of this paper is placed in the discussion section (4.2 Strengths and limitations and 4.3. Comparison with existing literature), it is better to put it in the front of the paper to make readers easier for understanding the starting point and value of the study;

IV.Others

1. The two statements about informed consent (L98 and L401) are different from each other;

2. A data extract (L184–185, "For home visits, it [VC] saves time. If I had to do a home visit, [I'd do] maybe two in half a day. Now I can see four people in that slot") needs to be italicized;

3. Repetition or Missing of Words, “……under the under the Mental Health Act (NHS, 1983)” (L193); “Both GPs (and psychiatrists?) described the presence……” (L222);

4. The citation format (including titles) of a few references needs to be checked and revised (L409–494). 

Author Response

Reviewer 1:

Response I & II : Thank you for this feedback. We have taken it on board and restructured the results on the newly submitted draft manuscript so that the link to our research question is clearer (all sub themes are now in the same hierarchy - they are either a challenge or a benefit so we hope this provides better clarity for the reader). To address your feedback regarding the naming standards of the themes, we have re-examined and added an extra theme which we hope allows for better organisation and co-ordination. 

Response III
1.    It is not appropriate to take the demographic characteristics of the participants as the results or findings of the thematic analysis, which can be placed in the section of 2.2. To address this, we have moved the description of patient demographics and the table to section 2.2.

  1.    Can the two tables (Table 2 and Table 3) be combined? It could reflect the professional background described. Can the participants’ codes be included in the table? Tables 2 and 3 have been merged. We have not added patient codes to ensure participant confidentiality is not potentially breeched.  

  2. The literature review of this paper is placed in the discussion section (4.2 Strengths and limitations and 4.3. Comparison with existing literature), it is better to put it in the front of the paper to make readers easier for understanding the starting point and value of the study. We have expanded on the paragraph in the introduction to better highlight the existing UK literature on the topic and make it easier for readers to understand the value of the study. However, we haven't moved all of the literature review to the introduction as a different reviewer wanted us to expand on the literature in the discussion. 

IV.Others
1. The two statements about informed consent (L98 and L401) are different from each other; The informed consent statement has been amended. 
2. A data extract (L184–185, "For home visits, it [VC] saves time. If I had to do a home visit, [I'd do] maybe two in half a day. Now I can see four people in that slot") needs to be italicized; This has been italicized. 
3. Repetition or Missing of Words, “……under the under the Mental Health Act (NHS, 1983)”(L193); “Both GPs (and psychiatrists?) described the presence……” (L222); This has been amended. 
4. The citation format (including titles) of a few references needs to be checked and revised (L409–494). The format of all references have been checked and edited.

Reviewer 2 Report

Interesting qualitative study on video consultations for diagnosis and follow-up in mental health. I consider that the manuscript is properly constructed, it has coherence and the methodology is adequate, but the discussion (specifically in comparison with existing literature) seems short to me and I would like if you could extend it two paragraphs and add more bibliography.

Author Response

Reviewer 2

Thank you for your feedback. We have expanded on the discussion and included more references.

Reviewer 3 Report

This is a well-written paper that describes the perspectives of psychiatrists and general practitioners regarding their experiences using videoconferencing for evaluating patients with mental health issues. This is important information given that remote healthcare visits are and will continue to become more frequent due to a number of reasons.

Strengths:

  • Methods are explained very nicely, particularly how data analysis was performed
  • Adequate sample size for qualitative work
  • Mix of urban and suburban participants (although mostly urban for psychiatrists)
  • Logical identification of three major themes and subthemes
  • Examples of quote from which themes/subthemes were derived seem appropriate
  • Limitations are described appropriately
  • Although there are some limitations that may decrease the generalizability, for the most part the findings will be generalizable and useful to a wide audience

Weaknesses:

  • Weaknesses are minor
  • Figure 1 is very blurry and cannot be read. This will need to be revised.
  • Line 222 – should this say “Both GPs and psychiatrists…”?
  • Line 403 references an appendix. This reviewer did not have access to that, but it is assumed it will be available to all upon publication

Author Response

Reviewer 3

Thank you for the feedback. The following points have been addressed:

  • Figure 1 is very blurry and cannot be read. This will need to be revised. This has been revised and a new figure uploaded.
  • Line 222 – should this say “Both GPs and psychiatrists…”? This has been edited.
  • Line 403 references an appendix. This reviewer did not have access to that, but it is assumed it will be available to all upon publication. This has been noted and the manuscript edited to say ‘The data presented in this study can be made available on request from the corresponding author.’

Round 2

Reviewer 1 Report

Thank you for the revision.